

# The margin of stability is affected differently when walking under quasi-random treadmill perturbations with or without full visual support

Zhuo Wang[1], Haoyu Xie[2] and Jung H. Chien[3]

[1] Department of Rehabilitation Medicine, West China Hospital, Sichuan University, Chengdu, Sichuan, China
[2] Department of Health & Rehabilitation Science, College of Allied Health Professions, University of Nebraska Medical Center, Omaha, NE, United States of America
[3] Independent Researcher, Omaha, NE, United States of America

Corresponding author
Jung H. Chien,
doghome6211@gmail.com

## ABSTRACT

**Background**. Sensory-motor perturbations have been widely used to assess astronauts' balance in standing during pre-/post- spaceflight. However, balance control during walking, where most falls occur, was less studied in these astronauts. A study found that applying either visual or platform oscillations reduced the margin of stability (MOS) in the anterior-posterior direction (MOSap) but increased MOS in the medial-lateral direction (MOSml) as a tradeoff. This tradeoff induced an asymmetric gait. This study extended the current knowledge to investigate overall stability under unpredictable environments. This study aimed to determine (1) whether quasi-random treadmill perturbations with or without full vision support would result in a significant reduction in MOSap but an increase in MOSml and (2) regardless of whether vision support was provided, quasi-random treadmill perturbations might result in asymmetric gait patterns.

**Methods**. Twenty healthy young adults participated in this study. Three experimental conditions were semi-randomly assigned to these participants as follows: (1) the control condition (Norm), walking normally with their preferred walking speed on the treadmill; (2) the treadmill perturbations with full vision condition (Slip), walking on the quasi-random varying-treadmill-belt-speeds with full vision support; and (3) the treadmill perturbations without full vision condition (Slip_VisionBlocked, blackout vision through customized vision-blocked goggles), walking on the quasi-random varying-treadmill-belt-speeds without full vision support. The dependent variables were MOSap, MOSml, and respective symmetric indices. A one-way repeated ANOVA measure or Friedman Test was applied to investigate the differences among the conditions mentioned above.

**Results**. There was an increase in MOSap in Slip ($p = 0.001$) but a decrease in MOSap in Slip_VisionBlocked ($p = 0.001$) compared to Norm condition. The MOSml was significantly greater in both Slip and Slip_VisionBlocked conditions compared to the Norm condition ($p = 0.011$; $p < 0.001$). An analysis of Wilcoxon signed-rank tests revealed that the symmetric index of MOSml in Slip_VisionBlocked ($p = 0.002$) was greater than in the Norm condition.

**Conclusion**. The novelty of this study was to investigate the effect of vision on the overall stability of walking under quasi-random treadmill perturbations. The results

revealed that overall stability and symmetry were controlled differently with/without full visual support. In light of these findings, it is imperative to take visual support into consideration while developing a sensory-motor training protocol. Asymmetric gait also required extra attention while walking on the quasi-random treadmill perturbations without full vision support to maintain overall stability.

## INTRODUCTION

Humans have long dreamed of landing on different planes throughout history as part of exploring the unknown. In 1972, the National Aeronautics and Space Administration (NASA) launched Apollo 17 to fulfill this dream. On the Moon, one of the astronauts, Dr. Harrison Schmitt, fell on the ground several times and required some clumsy assistance to get up (https://www.youtube.com/watch?v=Ke65jU_yYso, accessed on July 26, 2023). Although this astronaut was well-trained and in good health, his balance loss seemed inevitable and would cause further severe injuries. This fall probably was caused by a dim environment, sensory conflicts, or both (*Demir & Aydın, 2021*). Thus, to maintain balance, enhancing the ability to identify reliable sensory inputs under unpredictable environments becomes a critical topic; therefore, NASA specifically stressed the importance of adaptability training programs (https://www.nasa.gov/hrp/5-hazards-of-human-spaceflight, accessed on July 26, 2023). The fundamental concept of adaptability training involved challenging astronauts' sensory systems and forcing the central nervous system to identify a relatively reliable sensory system to maintain balance in unrehearsed and untrained situations. For instance, *Ozdemir et al. (2018)* measured balance control through a sensory organization test (SOT) in standing astronauts immediately after spaceflight, which required them to stand on a firm surface blindfolded (SOT condition 2, which identified the function of the somatosensory system) and stand on a sway referenced support surface blindfolded (SOT condition 5, which identified the function of the vestibular system) and SOT condition 1 (baseline condition, with eye-opened and standing on the solid surface). Based on the findings, (1) astronauts demonstrated larger center of pressure sway in post- than in pre-spaceflight in SOT 5, indicating that spaceflight led to the compromised balance control as a result of a lack of gravity; and (2) balance control was worsened in post-spaceflight when in SOT condition 5 than SOT condition 2, inferring that spaceflight had less impact on somatosensory function than the vestibular system. *Tays et al. (2021)* applied SOT condition 5 to measure the recovery in sensorimotor control after a long duration of spaceflight (approximately six months) in 15 astronauts. This previous study found that staying in microgravity for a long duration required at least 30 days for a full recovery in sensorimotor control. The above-mentioned measures were based on the sensory reweighting hypothesis (*Peterka, 2002*), which indicated that each sensory system was presented by a sensory channel consisting of a weighting factor that represents its

relative importance. In this regard, when a weighting factor was decreased for one sensory channel, a weighting factor would increase for another sensory channel. Thus, providing the unstable ground and blocking the visual system in the above-mentioned studies may be used to determine the roles of vestibular function on balance control in standing. However, most of the abovementioned studies focused primarily on balance control while standing rather than balance control while walking, where falls were most likely to occur. This study's main objective was to better understand how walking under sensory-conflicted conditions affected gait stability, which would be crucial for astronauts when encountering an unknown environment in space or other planes.

There have been several studies that utilize the above-mentioned sensory-manipulated paradigms to measure dynamic balance control during walking by measuring spatial gait parameters and the variability (*O'Connor & Kuo, 2009*; *McAndrew, Dingwell & Wilken, 2010*; *Chien et al., 2014*; *Roeles et al., 2018*; *Madehkhaksar et al., 2018*), margins of stability (MOS, *McAndrew Young, Wilken & Dingwell, 2012*; *Roeles et al., 2018*; *Madehkhaksar et al., 2018*, active control in gait (*Hu & Chien, 2021*), and gait symmetry (*McAndrew Young & Dingwell, 2012*). Could the alternations in gait characteristics and their variability be interpreted as changes in dynamic balance control during walking? *Pai & Bhatt (2007)* provided a definition that ''stability'' was the ability to maintain balance following an externally imposed perturbation or even during voluntary movement without requiring the change of the base of support (not global stability). Thus, it would appear that the margin of stability (MOS) might be a more practical method of measuring instantaneous gait stability induced by sensory perturbation than the gait characteristics. By integrating the center of mass (COM) velocity component into the margin of stability (MOS) measure, which was not taken into account in conventional gait characteristics measures, *Hof, Gazendam & Sinke (2005)* defined MOS as the ability to tolerate deviations from COM before losing balance during standing. As part of the calculation of MOS, the trunk component was also considered, while other gait measures only included the lower extremities. MOS could be calculated both in the medial-lateral (MOSml) and anterior-posterior (MOSap) directions. The deviation from the straight line of walking would be represented by a negative MOSml. As a result of this negative MOSml, a crossover step would be taken to prevent the human walking from falling to the side. It is not the same story for the MOSap. *Curtze, Buurke & McCrum (2023)* particularly emphasized that walking in the anterior-posterior direction is unstable because the act of walking has been described as ''controlled falling'' (*O'Connor & Kuo, 2009*). In other words, when one takes a step, the body leans forward and falls slightly, causing it to become unbalanced; then, one needs to catch the weight with one's outstretched foot to move the body back into balance. Therefore, the MOS should be negative where the extrapolated center of mass (XCOM) exceeds the center of pressure (COP) at the heel contact using the equation: MOSap = BOS (base of support) –XCOM.

The above-mentioned MOS method has also been widely used to identify the stability in pathologic gait in patients with stroke (*Tisserand et al., 2018*; *Kao et al., 2014*), patients with spinal cord injury (*Day et al., 2012*) and patients with Parkinson's disease (*Martelli et al., 2017*; *Moreno Catalá, Woitalla & Arampatzis, 2016*). In short, most of these studies found that the pathological groups had less stable MOS (smaller MOS values) in the AP

direction but more stable MOS (greater MOS values) in the ML direction, inferring that the pathological groups required a higher level of active control the COM movement in the ML direction than in the AP direction (*Bauby & Kuo, 2000*). It is worth clarifying that the MOS should not be interpreted as an indicator of global stability since no loss of balance or falls are reported in all the above-mentioned studies. Instead, the MOS should be interpreted as an instantaneous stability of body configuration (*Curtze, Buurke & McCrum, 2023*).

*McAndrew, Dingwell & Wilken (2010)* and *McAndrew Young & Dingwell, 2012* applied a pseudo-random platform and visual oscillations (the oscillation was a combination of four frequencies of sinusoidal waves) in both AP and ML directions on twelve healthy young adults to measure their gait stability during three-minute treadmill walking. These two previous studies observed that (1) both platform and visual oscillations increased step width and MOSml; however, these oscillations, regardless of the direction and type, decreased step length and MOSap; and (2) in terms of gait and MOS variability, using a platform and visual oscillations increased the variabilities regardless of the direction. As mentioned above, these results revealed that healthy young adults used conservative walking patterns, like the above-mentioned pathological group (shorter and wider steps), resulting in smaller MOSap and greater MOSml while encountering the oscillations. However, such observations were not found in *Madehkhaksar et al.*'s (*2018*) study. It was observed that stride length and MOS significantly decreased in the AP direction only when the oscillation moved to the right (dominant leg side, platform moved 12.8 centimeter with acceleration of 1.5 m/s$^2$), but not when the oscillations moved forward (accelerated the speed from 1.11 m/s toward 2.5 m/s), backward (decelerated the speed from 1.11 m/s toward 0 m/s), or to the left (non-dominant leg side) in the study (*Madehkhaksar et al., 2018*). In contrast, another study (*Roeles et al., 2018*) found that the platform oscillation only affected the MOS and stride length while these oscillations moved backward (dropping 40% of the original walking speed in 0.4 s) and toward the dominant leg side (5-centimeter platform translation in approximately 0.7 s) in both healthy young and older adults. Also, while visual capacity was reduced to <1 lx in 5 s, the MOSap was not affected compared to walking normally (*Roeles et al., 2018*), indicating that young and older adults could keep the same pace even in low light intensity. These inconsistent results might be attributed to (1) the body segments involved for the center of mass calculation, (2) the definition of the boundary of the base of support (toe or heel) for MOS calculation, and (3) the treadmill speed is not taken into consideration for MOS calculation. It should be noted that the equation used to calculate the XCOM in *McAndrew Young & Dingwell (2012)*, *Madehkhaksar et al. (2018)*, and *Roeles et al.*'s (*2018*) studies was the COM position plus its velocity divided by $\sqrt{g/l}$, where g was the acceleration of gravity and l was the distance from the marker on foot to the COM at the instant of heel strike. Accordingly, the MOSap in these studies (*McAndrew Young & Dingwell, 2012*; *Madehkhaksar et al., 2018*; *Roeles et al., 2018*) does not include the treadmill velocity component in the calculation of the XCOM, thereby causing concern about the validity of this XCOM calculation (*Curtze, Buurke & McCrum, 2023*).

*McAndrew Young & Dingwell (2012)* further demonstrated that these platform and visual oscillations caused the asymmetric MOS, a significantly greater MOS in the dominant leg

than in the non-dominant leg. This previous study indicated that these healthy young adults used the dominant side to control gait stability. These above-mentioned studies revealed a critical point: different amplitude and frequency combinations induced different impacts on gait characteristics and MOS in the AP and ML directions. As a part of NASA's mission to Mars, a combination of different amplitudes and frequencies in sensory manipulation paradigms would be more effective for identifying and further training astronauts to control their gait stability under unanticipated circumstances. With the help of a novel locomotor sensory organization test (LSOT), LSOT condition 4 (walking under treadmill perturbations with full vision support) and 5 (walking under treadmill perturbations without full vision support by wearing blackout goggles, (*Chien et al., 2014*; *Hu & Chien, 2021*), our team was able to specifically investigate the role of visual and vestibular function under quasi-random treadmill perturbations. Our team has demonstrated that the quasi-random treadmill perturbations significantly increased the variability in the length and width of the steps in both young and older adults (*Hu & Chien, 2021*), similar to the findings of *McAndrew Young & Dingwell (2012)* by quasi-randomly altering the acceleration/deceleration of treadmill speed. This quasi-random treadmill perturbation, however, did not affect the step length or step width in either young or older adults, regardless of visual support (*Hu & Chien, 2021*), emphasizing the importance of adjustment from step to step. Besides, the upper body movement and body moving velocity need to be considered to understand gait stability further.

Therefore, the main objective of this study was to investigate the overall stability of healthy young adults by measuring the MOS under treadmill perturbations with/without full visual support (LSOT conditions 1, 4, and 5; *Chien et al., 2014*) to make future comparisons with astronauts by measuring the spatial gait parameters, dynamic stability (MOS), and symmetry indices based on many studies mentioned above. Specifically, NASA supported this study as a pilot study for identifying sensorimotor responses under unexpected situations in the future. This study hypothesized that (1) the implementation of quasi-random treadmill perturbations would result in a decrease in MOSap but an increase in MOSml regardless of whether vision was provided or not; (2) greater decreases in MOSap and greater increases in MOSml would be observed while walking on quasi-random treadmill perturbations without full visual support than with visual support; and (3) quasi-random treadmill perturbations might result in asymmetric gait patterns regardless of whether vision support was present or not.

## METHODS

### Participants

A total of twenty healthy young adults participated in this study. A summary of the demographic characteristics of all participants is presented in Table 1. A variety of advertising methods were employed, including flyers posted on university campuses and in local community centers, as well as an online bulletin posted on the university's website to recruit these healthy young adults. All young adults were required to meet the following inclusion criteria: (1) participants must be between the ages of 18 and 30 years

old, (2) all participants must be free of musculoskeletal deficits and have no history of extremity injuries, (3) participants must not have any joint surgeries that would affect their gait pattern, and (4) participants must pass the dizzy handicap inventory (score = 0), indicating that potential vestibular dysfunction may not exist. All surveys were completed on the day of data collection. The University of Nebraska Medical Center's medical ethics committee (IRB# 340-10-FB) approved this study in accordance with relevant guidelines and regulations. An informed consent form was provided to each participant one week prior to the collection of data. On the day of data collection, the experiment would only begin if participants signed the informed consent form voluntarily. Data was collected between April 18, 2013, and April 17, 2014. The statistical power was estimated using the following two methods. Firstly, we evaluated the power from our previous study (*Chien et al., 2014*), which investigated the changes in the net center of pressure while ten healthy young adults walked under six sensory-conflicting conditions (three conditions were also used in the current study). According to this previous study (*Chien et al., 2014*), a one-way repeated ANOVA measure was applied to compare the net center of pressure differences among these six conditions. As a result of recruiting only ten healthy young adults, the partial eta squared value was 0.544, indicating the large effect size (*Cohen (1988)*, $\eta2$ value for a large effect size was 0.14, for a moderate effect size was 0.06; for a small effect size was 0.01). The second method was to use the power estimate software G∗ power (http://www.gpower.hhu.de/) to estimate the statistical power. F-tests, ANOVA: Repeated measures within factors, A priori: Compute the required sample size given power, alpha, and effect size, Partial eta-squared ($\eta2 = 0.14$), group = 1 and measurement = 3 were selected in the software G∗ power. According to the estimation, recruiting 18 healthy young adults could achieve 95% power when using the repeated measure in the present study. In short, recruiting a total of twenty healthy young participants should have sufficient power.

**Experimental protocol**

The experimental protocol was also adapted based on the previous study (*Hu & Chien, 2021*). Before data collection, each participant's anthropometric data were measured. Each participant was asked, "Which leg do you prefer to kick a ball?" to identify the dominant leg. Next, each participant's preferred walking speed (PWS) would be recognized as follows. Participants stood on the side of the treadmill. The experimenter speeded up the treadmill speed to 0.8 m/s. Then, participants stepped and continuously walked on the treadmill. After 20 s, the experimenter asked participants, "Is this speed like your comfortable walking speed, such as walking around the neighborhood?" Depending on the responses from participants, the treadmill speed was increased/decreased by 0.1 m/s by the experimenter. This procedure was repeated until the PWS was verified. Once the PWS was confirmed, participants walked on this PWS for another five-minute walk to familiarize themselves with the treadmill. After a five-minute familiarization, a two-minute mandatory rest was given to each participant. Then, participants were provided three two-minute locomotor tasks (Fig. 1A): Norm, Slip, and Slip_VisionBlocked conditions. Two quasi-random treadmill perturbations with (Slip)/without (Slip_VisionBlocked) full vision support and one baseline (Norm) locomotor task were assigned to twenty

**Table 1  Participant's information.**

| | Age (Years) | PWS (m/s) | Gender | Weight (kg) | Height (cm) |
|---|---|---|---|---|---|
| Sub01 | 20 | 1.50 | M | 81.65 | 190 |
| Sub02 | 28 | 1.40 | M | 86.18 | 177 |
| Sub03 | 22 | 1.40 | F | 49.89 | 163 |
| Sub04 | 19 | 1.40 | F | 80.72 | 163 |
| Sub05 | 20 | 1.50 | M | 74.55 | 170 |
| Sub06 | 24 | 1.40 | M | 85.23 | 173 |
| Sub07 | 24 | 1.20 | F | 54.56 | 164 |
| Sub08 | 21 | 1.40 | F | 66.82 | 173 |
| Sub09 | 29 | 1.30 | M | 70.46 | 177 |
| Sub10 | 20 | 1.60 | F | 63.63 | 170 |
| Sub11 | 21 | 1.60 | M | 53.00 | 170 |
| Sub12 | 22 | 1.60 | F | 75.00 | 173 |
| Sub13 | 27 | 1.30 | F | 51.00 | 159 |
| Sub14 | 20 | 1.60 | F | 53.00 | 167 |
| Sub15 | 22 | 1.40 | F | 63.00 | 155 |
| Sub16 | 21 | 1.50 | M | 79.00 | 180 |
| Sub17 | 24 | 1.60 | M | 86.18 | 185 |
| Sub18 | 23 | 1.30 | M | 58.00 | 167 |
| Sub19 | 22 | 1.40 | F | 58.00 | 160 |
| Sub20 | 21 | 1.40 | M | 74.00 | 173 |
| Avg. | **22.55** | **1.44** | **10M/10F** | **68.19** | **170.45** |
| SD. | **2.78** | **0.12** | | **12.68** | **8.74** |

**Notes.**
Sub, participants number; M, male; F, female; PWS, preferred walking speed.

healthy young participants (Fig. 1B). For the Norm condition, participants walked on a fixed treadmill speed (their PWS) with full vision support. For the Slip condition, participants walked on quasi-random treadmill perturbations with full vision support. For the Slip_VisionBlocked condition, participants walked on quasi-random treadmill perturbations without full vision support. Firstly, twenty healthy adults were semi-randomly divided into two groups of ten individuals, each using the random number generation method embedded in the MATLAB 2021b commercial engineering software (https://www.mathworks.com/help/matlab/ref/rand.html, MathWorks Inc., USA). In group #1, the Slip condition was always assigned before the Slip_VisionBlocked condition. Also, the Slip_VisionBlocked condition was always assigned before the Slip condition for group #2. According to this experimental design, there was a counterbalance between the patterns of two different treadmill perturbation sequences (Fig. 1B, Slip and Slip_VisionBlocked, *Hu & Chien, 2021*). After each participant completed each condition, they were asked, "What are your feelings now?" The experimenters wrote down a note based on each participant's description.

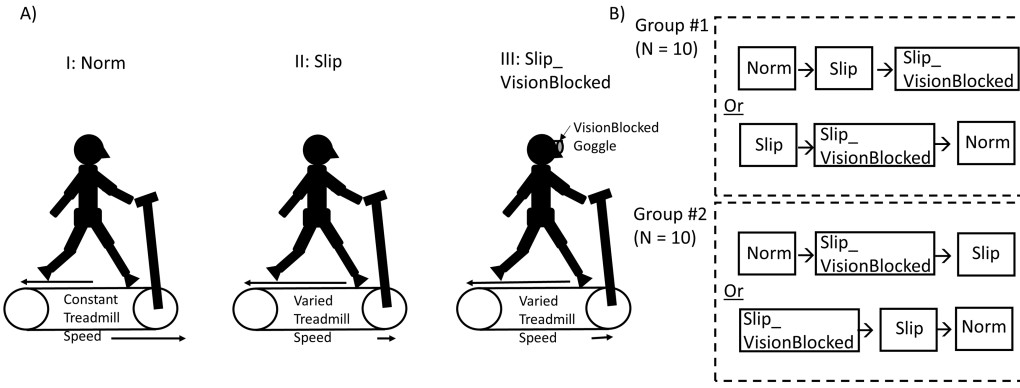

**Figure 1 Experimental Paradigm and Group Assignment.** (A) Two quasi-random treadmill perturbations with (Slip)/without (Slip_VisionBlocked) full vision support and one baseline (Norm) locomotor task. For the Norm condition, participants walked on a fixed treadmill speed (their PWS) with full vision support. For the Slip condition, participants walked on quasi-random treadmill perturbations with full vision support. For the Slip_VisionBlocked condition, participants walked on quasi-random treadmill perturbations without full vision support. (B) Twenty healthy adults were randomly divided into two groups of ten individuals. In group #1, the Slip condition was permanently assigned before the Slip_VisionBlocked condition. Moreover, the Slip_VisionBlocked condition was permanently assigned before the Slip condition for group #2. When wearing VisionBlocked goggles, light intensity could be reduced from approximately 150 lx (light intensity in a regular room) to approximately 0.7 lx (full moon on a clear night when walking on a dark street without a streetlight).

## Equipment setup

The body landmark trajectories were captured by a motion capture system (Optotak Certus, Northern Digital Inc, Waterloo, Canada). This motion capture system contained two motion capture cylinders with three lenses for each cylinder, placed two meters away from the treadmill on the side with a capturing rate of 100 Hz. This two-meter distance was the optimal distance for the system to fully capture the marker trajectories (https://tsgdoc.socsci.ru.nl/images/e/eb/Optotrak_Certus_User_Guide_rev_6%28IL-1070106%29.pdf). COM position was calculated based on segmental positions, including pelvis, thighs, shanks, feet, seventh cervical vertebra (C7), tenth thoracic vertebra (T10), jugular notch, xiphoid process, anterior superior iliac spine, posterior superior iliac spine, greater trochanter, medial femoral epicondyle, lateral femoral epicondyle, thigh, tibia, medial malleolus, lateral malleolus, posterior calcaneus, medial calcaneus, lateral calcaneus, heel, 2nd (M2) and 5th (M5) metatarsal head (*Vanrenterghem et al., 2010*). This model was also suggested by *Havens, Mukherjee & Finley (2018)* to be the best-simplified model to represent the whole-body COM kinematics due to its relatively small errors generated, specifically for calculating the margin of stability.

A customized visual basic script (Microsoft, Redmond, WA, USA) was embedded into Bertec Device SDK to generate these two quasi-random treadmill perturbations. These two quasi-random treadmill perturbations were designed as follows (Fig. 2): Step #1) to generate time blocks—continuously and randomly generated a value (between 5–10 s) as a time block until the sum of these values reached 120 (120 s); Step #2) to generate the preferred walking speed (PWS) blocks–generated a value between −20 to 20

(positive value indicates acceleration, a negative value indicates deceleration). Next, this value was assigned to the time blocks and added to the already generated values (PWS at zero seconds). Specifically, the value in each time bock cannot be over either −20 or 20; otherwise, step #2 was redone. For the purpose of comparing the differences between these two sequences, the total amounts of frequencies and amplitudes of quasi-random treadmill perturbations were kept the same. The detail is shown in Fig. 2. For the Slip_VisionBlocked condition, participants walked on the treadmill perturbation and wore blackout goggles with a layer of 5% car-tinting vinyl. Also, the peripheral vision was blocked by the goggles. This study permitted a small amount of light to penetrate the blackout goggles compared to completely blindfolding. When wearing these goggles, light intensity could be reduced from approximately 150 lx (light intensity in a regular office) to approximately 0.7 lx (full moon on a clear night when walking on a dark street without a streetlight). The light intensities were obtained by a light meter (Dr. Meter, support@drmeter.com) inside the goggles. The room light intensities were captured between trials to ensure consistency throughout the data collection. Participants were required to take a two-minute rest between trials to catch up on their breath between each trial. An instrumental treadmill (Bertec Corp., Columbus, OH, United States) with a sampling rate of 300 Hz was used. The embedded force plates in this instrumental treadmill were used to identify the critical gait events by synchronizing the motion capture data through the NDI's first principal software (Optotak Certus, Northern Digital Inc., Waterloo, Canada): heel-strike and toe-off of each leg. The definition of the heel strike was the first frame in which vertical force was over 10N and continuously lasted for 40 ms (*Chien et al., 2014*). The toe-off was the first frame in which vertical force was below 10N and continuously lasted for 40 ms.

## Data analysis

The dependent variables in the current study were the MOS in the anterior-posterior direction (MOSap), MOS in the medial-lateral direction (MOSml), MOS variability in the anterior-posterior direction (MOSVap), MOS variability in the medial-lateral direction (MOSVml), step length, step length variability, step width, step width variability, symmetric index of MOSap (MOSSIap), and symmetric index of MOSml (MOSSIml). The MOS (Fig. 3) was initially introduced by Hof et al.'s study (2005). The position of COM was calculated by body landmarks with low-pass filters using a 4th-order Butterworth filter with a cut-off frequency of 6 Hz using visual 3D (C-Motion Inc., Rockville, MD, USA). Then, the XCOM was defined as

$$XCOM = x + \frac{\dot{x} + \dot{x}_{treadmill}}{\omega_0}$$

Where $x$ was the COM position, $\dot{x}$ was the velocity of COM, and $\dot{x}_{treadmill}$ was the velocity of the treadmill belt velocity (*Süptitz et al., 2012*). The $\dot{x}$ was the first derivative of the COM position, and the $\dot{x}_{treadmill}$ was the distance divided by the time between two consecutive heel contacts. The $\omega_0 = \sqrt{\frac{g}{L}}$, where $g = 9.81$ m/s$^2$, and L was the distance from COM to the heel at the moment of heel strike. The BOS was defined using the heel position at the moment of heel strike (*Süptitz et al., 2012*; *Fallahtafti et al., 2021*; *Lu, Xie & Chien, 2022*;

**Steps to develop varying treadmill speed sequences:**

**Step #1 Time-interval blocks generation**

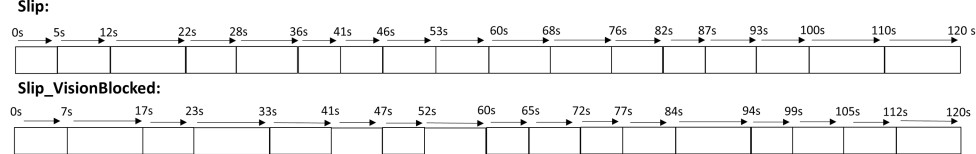

**Step #2 Number of varying speeds generation and assignment**

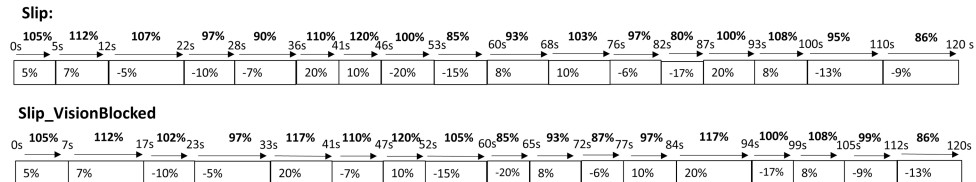

**Figure 2 Perturbations of treadmill.** (A) Step #1) Time-interval blocks generation: to generate time block continuously and randomly generated a value (between 5–10 s) as a time block until the sum of these values reached 120 (120 s). There were 17 time blocks in each perturbations. For instance, 5s –> 12s meant that this time block lasted for 7 s and so on. (B) Step #2) Number of varying speeds generation and assignment: generated a value between −20 to 20 (a positive value indicates acceleration, a negative value indicates deceleration). Next, this value was assigned to the time blocks and added to the already generated values (PWS at zero seconds). Specifically, the value in each time bock cannot be over either −20 or 20; otherwise, step #2 was redone. For instance, 7% meant that plus 7% of preferred walking speed in the preferred walking speed in previous time block, such as if the previous preferred walking speed was 105% than the new preferred walking speed in the current time block was 112%, and so on.

*Roeles et al., 2018; Kao et al., 2014; Punt et al., 2017; Peebles et al., 2017; Martelli et al., 2017; Curtze, Buurke & McCrum, 2023;* Fig. 3).

In the anterior-posterior direction:

MOSap = BOS (coordinate of heel of leading leg in the anterior-posterior direction) –XCOMap.

In the medial-lateral direction:

Right Leg:

MOSml = M5 (coordinate of M5 of leading leg in the medial-lateral direction) - XCOMml

Left Leg:

MOSml = XCOMml - M5 (coordinate of M5 of leading leg in the medial-lateral direction)

The symmetric index of MOSap and MOSml, the symmetric index was calculated by

$$Symmetric\ Index = \frac{Y_D - Y_{ND}}{0.5(Y_D + Y_{ND})}$$

Where $Y$ was the value of MOSap or MOSml, $D$ was the dominant leg, and $ND$ was the non-dominant leg (*Kaczmarczyk et al., 2017*). If the symmetric index (MOSSIap, MOSSIml) approached zero, MOS tended to be symmetric. If the symmetric index was positive, the asymmetric patterns tended to the dominant leg and vice versa. The gait variability was defined as the coefficient of variance, which was effectively a normalized

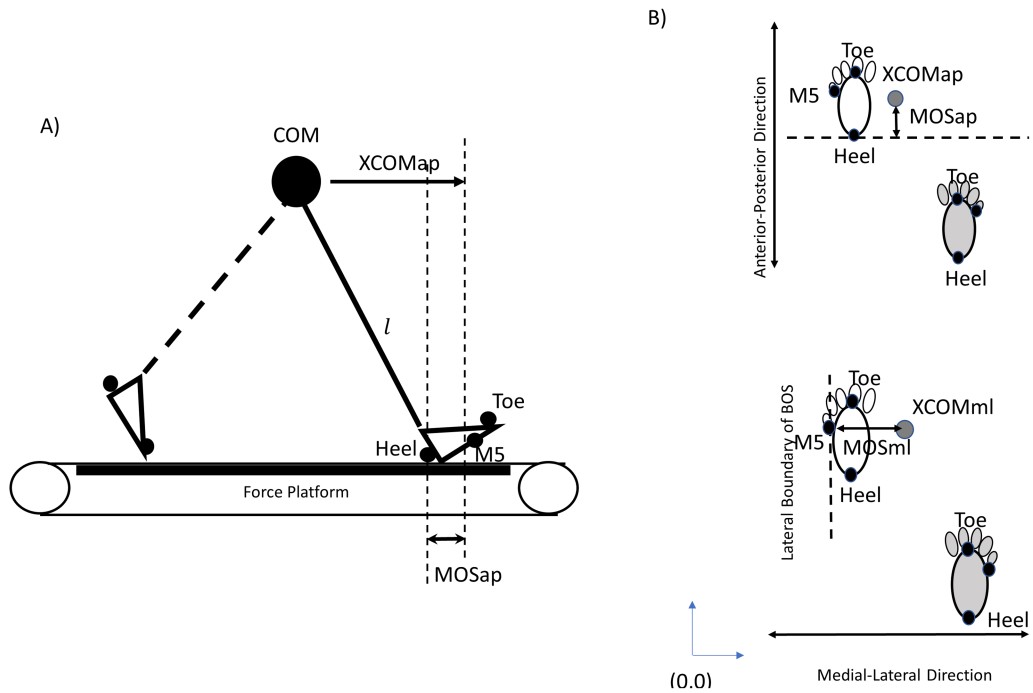

**Figure 3  The calculation of MOS.** (A) An inverted pendulum model shown at the moment of heel strike could be used to calculate the margin of stability (MOS) in the anterior-posterior direction. (B) The BOSap (Base of Support in the anterior-posterior direction) was defined as an anterior boundary limit of BOS, which was the coordinate of heel marker at the moment of heel contact. Then, MOSap was estimated as the boundary limit of BOS in the anterior-posterior minus the XCOMap. The BOSml (Base of Support in the medial-lateral direction) was defined as a lateral boundary limit of BOS, which was the coordinate of M5 at the moment of heel strike. Therefore, for left leg, the MOSml was estimated as the XCOMml minus BOSml; however, for right leg, the MOSml was estimated as the BOSml minus XCOMml because the original point (0,0) was set in the left-rear corner. By this method, if the dynamic stability was considered relatively stable in the ML direction, the values of MOSml should be positive.

or relative measure of the variation in a data set in time series (standard deviation *100 / mean) because each participant might have different numbers of steps within a two-minute walking trial.

## Statistical analysis

A Shapiro–Wilk normality test was applied to this study to identify the normality of each dependent variable, with the alpha value set at 0.05. If the data were normally distributed, a one-way repeated ANOVA measure was applied to investigate the effect of the different sensory-conflicted situations on walking in step length, step width, step length variability, step width variability, MOSap, MOSml, MOSVap, MOSVml, MOSSIap, and MOSSIml. A Bonferroni *post hoc* correction was used for multiple comparisons. All statistical analyses were performed through SPSS 26 (IBM, USA). For Bonferroni correction, SPSS took the observed (uncorrected) *p*-value and multiplied it by the number of comparisons made (https://www.ibm.com/support/pages/calculation-bonferroni-adjusted-p-values, reviewed on June 30, 2023). In this present study, the uncorrected *p*-value was multiplied by

three (three pairs of comparisons in the current study). Then, if this corrected value was smaller than 0.05, the results concluded that the difference was significant. If the data were not normally distributed, a Friedman test was applied to identify the effect of different conditions on each dependent variable. Wilcoxon signed-rank test was used for pairwise comparisons. The effect size was calculated using the partial eta-squared method (*Cohen, 1988*). Based on *Cohen (1988)*, the value of 0.14 was for a large effect size, 0.06 was for a moderate effect size, and 0.01 was for a small effect size.

## RESULTS

### Normality tests

The results of the Shapiro–Wilk test revealed that the alpha value was greater than 0.05 for MOSap, MOSml, MOSVml, MOSSIap, step length, step width, step length variability, and step width variability, indicating the normal distribution. However, for MOSVap and MOSSIml, the alpha values were smaller than 0.05, showing a non-normal distribution.

### The effect of quasi-random treadmill perturbations with/without full vision support on margins of stability and its variability

A significant effect of quasi-random treadmill perturbation was found in the MOSap ($F_{2,38} = 58.646$, $p < 0.001$), in the MOSml ($F_{2,38} = 13.577$, $p < 0.001$), in MOSVap ($\chi^2 = 32.7$, $p < 0.001$) and in the MOSVml ($F_{2,38} = 67.409$, $p < 0.001$). As indicated by pairwise comparisons, the MOSap increased significantly in the Slip condition ($p < 0.001$) but decreased significantly in the Slip_VisionBlocked condition ($p < 0.001$) when compared to the Norm condition (Fig. 4A). Also, a significant increase in the MOSap in the Slip condition than in the Slip_VisionBlocked condition was found ($p < 0.001$, Fig. 4A). For the medial-lateral direction, MOSml was significantly greater in both Slip ($p = 0.011$) and Slip_VisionBlocked conditions ($p < 0.001$) compared to the Norm condition (Fig. 5A). For both MOSVap and MOSVml, the variabilities were significantly greater in both Slip ($p = 0.005$, $p < 0.001$, respectively) and Slip_VisionBlocked conditions ($p < 0.001$, $p < 0.001$, respectively) compared to the Norm condition (Figs. 4B and 5B). Also, the variabilities for both MOSVap and MOSVml in the Slip_VisionBlocked condition were significantly greater than in the Slip condition ($p < 0.001$, $p = 0.002$, respectively). The partial eta squared values were 0.755 for MOSap, 0.417 for MOSml, and 0.78 for MOSVml, indicating a large effect size. More details are shown in Figs. 4 and 5.

### The effect of quasi-random treadmill perturbations with/without full vision support on spatial–temporal gait parameters and respective variabilities

A significant effect of quasi-random treadmill perturbation was found in the step length variability ($F_{2,38} = 52.208$, $p < 0.001$), in the step width ($F_{2,38} = 18.462$, $p < 0.001$), and the step width variability ($F_{2,38} = 31.152$, $p < 0.001$). The pairwise comparisons revealed significantly greater values in step length variability ($p < 0.001$) and step width variability ($p = 0.02$) while walking in the Slip condition than walking in the Norm condition. Also, the pairwise comparisons revealed significantly greater values in step length variability

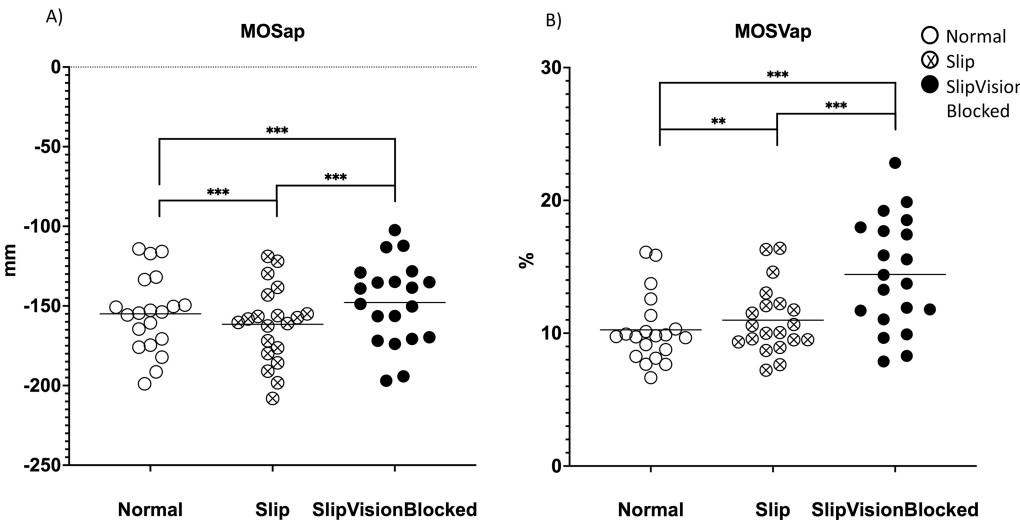

**Figure 4  Margins of stability (MOS) and its variability in the anterior-posterior direction.**
(A) MOSap–means value of MOS within each condition in the anterior-posterior direction. (B) MOSVap–coefficient of variance of MOS within each condition in the anterior-posterior direction. *: $p < 0.05$; **: $p < 0.01$; ***: $p < 0.001$. Each dot represented an each healthy participant. Horizontal line represented the mean value.

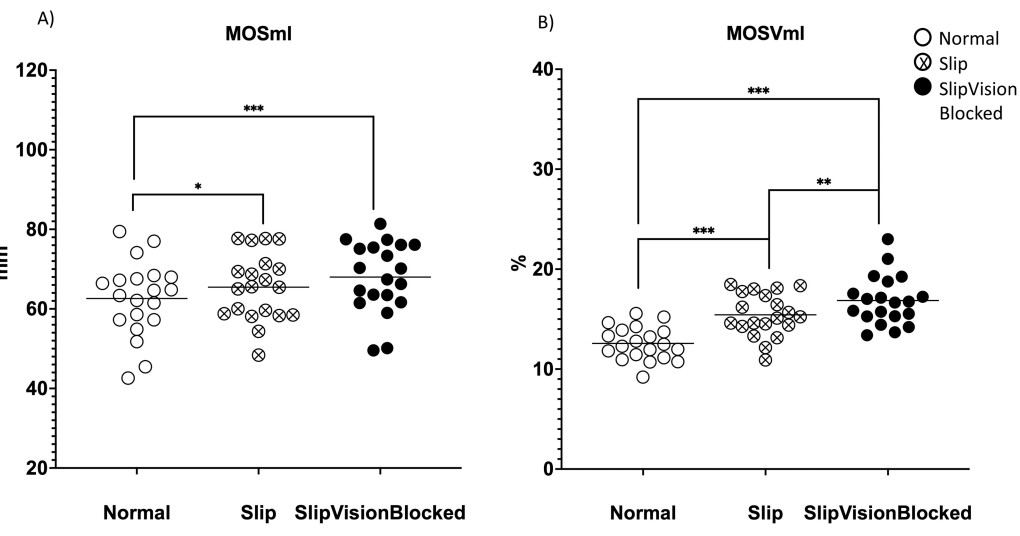

**Figure 5  Margin of stability (MOS) and its variability in the medial-lateral direction.** (A) MOSml–means value of MOS within each condition in the medial-lateral direction. (B) MOSVml–coefficient of variance of MOS within each condition in the medial-lateral direction. *: $p < 0.05$; **: $p < 0.01$; ***: $p < 0.001$. Each dot represented an each healthy participant. Horizontal line represented the mean value.

($p < 0.001$), step width ($p < 0.001$), and step width variability ($p < 0.001$) while walking in the Slip_VisionBlocked condition than walking in the Norm condition. The partial eta squared values were 0.73 for step length variability, 0.493 for step width, and 0.621 for step width variability, indicating a large effect size. More details are shown in Table 2.

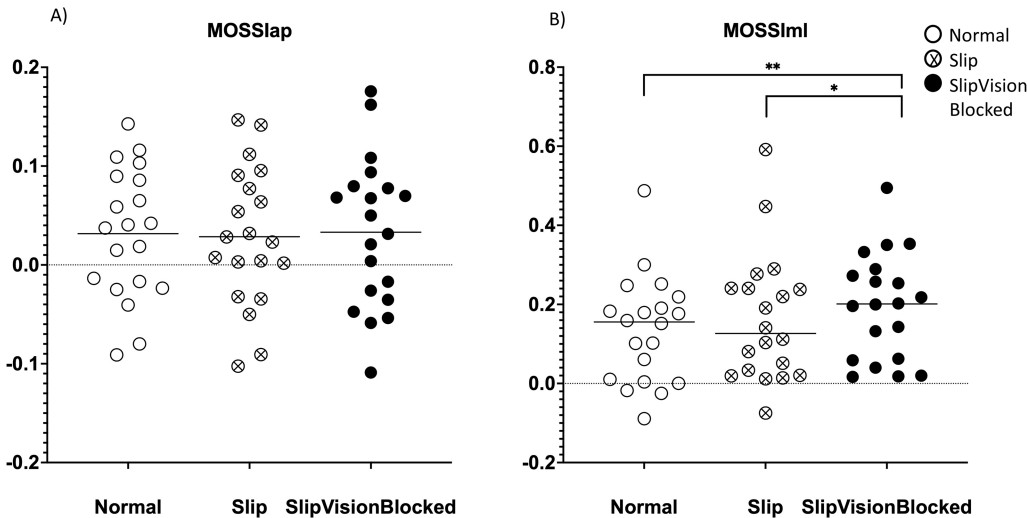

**Figure 6** **Margin of stability (MOS) symmetric index.** (A) MOSap symmetric index (MOSSIap)–MOS symmetric index in the anterior-posterior direction. (B) MOSml symmetric Index (MOSSIml)–MOS symmetric index in the medial-lateral direction. Higher vales meant greater deviation on an individual leg. Positive symmetric index values represented that participants tended to use the dominant leg. Close to zero indicated close to no deviation of MOS during gait. Horizontal line: the mean value.

**Table 2** **Spatial–temporal gait parameters and respective variabilities.** The one-way repeated ANOVA measure without adding preferred walking speed was used to investigate the effect of treadmill-induced perturbations with/without full vision support on step length, step length variability, step width, and step width variability. The *post-hoc* comparisons were corrected by Bonferroni correction.

| | Norm | Slip | Slip_Vision Blocked | Effect | Norm vs. Slip | Slip vs. Slip_Vision Blocked | Norm vs. Slip_Vision Block |
|---|---|---|---|---|---|---|---|
| Step length (mm) | 578.73 (68.27) | 572.71 (69.62) | 574.92 (75.24) | $F = 0.258, p = 0.775$ | NA | NA | NA |
| Step length variability (%) | 3.76 (0.88) | 7.33 (2.21) | 7.65 (1.36) | $F = 52.208, p < 0.001$ | $p < 0.001$ | NS | $p < 0.001$ |
| Step width (mm) | 116.81 (28.61) | 120.75 (24.48) | 130.01 (25.48) | $F = 18.462, p < 0.001$ | NS | $p < 0.001$ | $p < 0.001$ |
| Step width variability (mm) | 14.33 (2.76) | 16.00 (3.88) | 19.27 (3.62) | $F = 31.152, p < 0.001$ | $p = 0.02$ | $p = 0.001$ | $p < 0.001$ |

**Notes.**
Norm, the control condition, walking normally with their preferred walking speed on the treadmill;; Slip, the treadmill-induced perturbation with full vision condition, walking on the varied-treadmill-belt-speeds with full vision support; Slip_VisionBlocked, the treadmill-induced perturbation without full vision condition, walking on the varied-treadmill-belt-speeds without full visual support; NA, not available, the effect of different conditions did not reach the statistical significance; NS, not significant.
Data are shown as mean (standard deviation).

## The effect of quasi-random treadmill perturbations with/without full vision support on the symmetric index of margins of stability

The Friedman tests demonstrated that significant effects of quasi-random treadmill perturbations with/without full vision support were found in the symmetric index (SI) of MOSml ($\chi^2 = 11.1$, $p = 0.004$). Wilcoxon signed-rank tests demonstrated that the SI of MOSml in Slip_VisionBlocked condition ($p = 0.002$) was larger than Norm. Also,

the Wilcoxon signed rank test revealed that the SI of MOSml was larger ($p = 0.015$) in Slip_VisionBlocked than in the Slip condition (Fig. 6).

## DISCUSSION

The purpose of this study was to determine how healthy young adults adjusted their MOS under quasi-random treadmill perturbations with or without full vision support. It would be beneficial to investigate the above-mentioned questions to gain further insight into the development of astronaut sensorimotor training protocols in locomotion. As shown in the present study, quasi-random treadmill perturbations decreased the mean value of MOSap while increasing the mean value of MOSml in Slip_VisionBlocked condition compared to other conditions. As compared to the other conditions, treadmill perturbations increased the mean values of MOSap and MOSml in the Slip Condition when full vision support was present. When walking in Slip_VisionBlocked, the symmetric index in the ML direction (more asymmetric gait) increased significantly towards the dominant leg when compared with Norm and Slip conditions, suggesting that the mean value of MOSml was controlled in the dominant leg more than the non-dominant leg during treadmill perturbations without full vision support.

### MOS values in the current study and previous studies

A MOS method has been widely applied to measure instantaneous stability during walking in healthy and pathological individuals. The method for calculating the MOS used in the studies mentioned above in the introduction was inconsistent. Therefore, it was necessary to compare the results of this study with those of other studies utilizing similar calculation methods. *Curtze, Buurke & McCrum (2023)* strongly suggested that treadmill speed should be taken into account when determining the MOS for treadmill walking. It is also noteworthy that the MOS at heel contact should be negative due to the XCOM exceeding the BOS boundary throughout the gait cycle (coordinates of the heel marker in the anterior-posterior direction): MOSap = BOSap (heel)—XCOM (*Curtze, Buurke & McCrum, 2023*; *Lu, Xie & Chien, 2022*; *Fallahtafti et al., 2021*). This study found that the mean value of MOSap was negative, consistent with MOSap values found in previous studies when participants walked normally (Table 3). In contrast, the MOSap and MOSml in the study of *McAndrew Young & Dingwell (2012)* were much greater than those in the present study and the studies mentioned above (*Curtze, Buurke & McCrum, 2023*; *Roeles et al., 2018*; *Fallahtafti et al., 2021*). It was, therefore, imperative that additional attention be paid to the mean values of MOSap and MOSml among the studies discussed above under similar sensory-conflicted conditions, particularly when treadmill speed is not taken into account in calculating MOS, and only the pelvis is used to calculate the center of mass.

### Walking under quasi-random treadmill perturbation without full vision support decreased MOSap

Vision was considered to be the dominant sensory system during walking. Walking in 0.7 lx (full moon on a clear night when walking on a dark street without streetlight)
**Table 3  The MOS values from previous studies and this present study.**

| BOS (Heel) - XCOM | MOSap (with treadmill speed) | | | | |
|---|---|---|---|---|---|
| | Present Study | *Lu, Xie & Chien (2022)* | *Fallahtafti et al. (2021)* | *Curtze, Buurke & Mc-Crum (2023)* | *McAndrew Young & Dingwell (2012)* (without treadmill speed) |
| BOS - XCOM | Approximately -155 mm | Approximately-155 mm | Approximately -170 mm | between -100 to -200 mm | +350 mm |
| | MOSml (with treadmill speed) | | | | |
| | Present Study | *Roeles et al. (2018)* | *Fallahtafti et al. (2021)* | *Curtze, Buurke & Mc-Crum (2023)* | *McAndrew Young & Dingwell (2012)* (without treadmill speed) |
| | approximately 60 mm –M5 | approximately 60 mm –Lateral malleolus | approximately 35 mm –Heel | approximately 60 mm –Lateral malleolus  approximately 30 mm –Heel | approximately 80 mm –M5 |

environment increased the variabilities in net center of pressure trajectories (*Chien et al., 2014*), as well as the area of heel contact distribution within 200 steps (*Ren, Lin & Chien, 2022*), as compared to walking in 150 lx (the intensity of light in a regular office) in young adults. However, when young adults walked in near-darkness (5 lumens, equivalent to the amount of light output of five candles), no differences were observed in their spatial gait parameter as compared to walking normally (1000 lumens, a bright office, (*Naaman et al., 2023*). Also, when the environment suddenly became dark for 5 s, the MOSap within this timeframe did not appear to be affected in young adults while walking (*Roeles et al., 2018*). In short, the reduced visibility during walking on fixed surfaces appeared to increase the variability of gait but not the spatial gait parameter or MOSap. However, in the present study, a decrease in the mean value of MOSap was observed in Slip_VisionBlocked compared to the Norm condition. To explain this observation, as a first step, it was necessary to gain a basic understanding of MOS's fundamental components: the XCOM coordinate and the boundary of the BOS (MOSap = BOSap –XCOMap). A decrease in MOSap could have five possible combinations, according to the equation: (1) a decrease in BOSap but no change in XCOMap, (2) a decrease in BOSap but an increase in XCOMap, (3) no change in BOSap and an increase in XCOMap, (4) a decrease in BOSap and XCOMap, but BOSap decreased more than XCOMap, and (5) BOSap and XCOMap both increased, but XCOMap increased more than BOSap. *Süptitz et al. (2012)* further suggested that (1) BOSap and XCOMap increased or decreased when treadmill speed increased or decreased, but at different rates, and (2) XCOMap and BOSap both played an essential role in the changes in MOSap during treadmill speed changes. A study conducted by *McAndrew Young & Dingwell (2012)* found that decreases in MOSap were correlated with decreases in step length when walking under sensory-conflicted conditions, such as oscillations on the surface or in visual surroundings. Thus, the decreases in MOSap and

step length might represent a tradeoff when encountering sensory challenging conditions. It was possible, in the present study, that the decreases in the mean value of MOSap (mean value of the instantaneous stability in the AP direction at each heel strike within a condition) could be attributed to the implementation of treadmill perturbations without full visual support during walking. The interpretation of the results should, however, be approached cautiously due to the limitations of the experimental design (More details are described in limitation). It was worth noting that, in this Slip_VisionBlocked condition, the mean treadmill speed was 102% of the preferred walking speed, which was close to the preferred walking speed during normal walking. Also, no significant differences in step length were found between the Norm and Slip_VisionBlocked conditions in the present study. There was a possibility that the mean of step length was averaged out as a result of step-by-step adjustments associated with slowed-down and sped-up treadmill settings in Slip_VisionBlocked. As suggested by *Herssens et al. (2021)* study, no differences in step length may lead to no differences in BOSap between Norm and Slip_VisionBlocked conditions in this present study. Accordingly, the decrease in the mean value of MOSap might, therefore, be attributed to the increase in the mean value of XCOMap after step-to-step adjustments when comparing this Slip_VisionBlocked condition with the Norm condition (combination #5 above). According to *Süptitz et al. (2012)*, an increase in XCOMap was associated with increases in COM and COM velocity. Consequently, it is possible that walking on a treadmill under quasi-random treadmill perturbations without full vision support triggered responses that caused the mean value of COM to move farther and faster, thus overcounting the instability caused by treadmill perturbations and reducing visibility from an overall perspective.

## Walking under quasi-random treadmill perturbation with full vision support increased MOSap

As previously demonstrated with full vision support (*Süptitz et al., 2012*; *Hak et al., 2015*), when external conditions disrupted the steady state of MOS, the first response was to increase the BOSap in order to maintain stability. *Buurke et al. (2019)* further suggested that a treadmill perturbation not only increased the BOSap but also reduced the forward excursion of XCOMs to increase their MOSap as well as their stability. Thus, it was not surprising to observe that walking in the Slip condition significantly increased the mean value of MOSap compared to walking in the Slip_VisionBlocked condition, demonstrating that vision was essential to maneuvering stability during walking under perturbations. Firstly, based on the discussion in the previous section, an increase in MOSap (MOSap = BOSap - XCOMap) could possibly be (1) an increase in BOSap but no change in XCOMap, (2) an increase in BOSap but a decrease in XCOMap, (3) no change in BOSap and a decrease in XCOMap, (4) an increase in BOSap and XCOMap, but BOSap increased more than XCOMap, and (5) BOSap and XCOMap both decreased, but XCOMap decreased more than BOSap. The increases in the mean value of MOSap while walking in the Slip condition compared to walking in the Slip_VisionBlocked condition in the present study might, therefore, not be attributed to the increase in BOSap but the decrease in XCOMap (combination #2). Again, it must be emphasized that the averaging effect due

to the limitations of the experimental design cannot be ignored in the Slip condition. The mean treadmill speed was 99.3% of the preferred walking speed in this Slip condition. This was the rationale that there were no differences in step length between the Slip and Slip_VisionBlocked conditions. Similar to previous discussions, the decrease in the mean value of XCOMap may play a critical role in the increase in the mean value of MOSap after step-by-step adjustments when comparing the Slip condition with the Slip_VisionBlocked or Norm condition (combination #2 in this section). According to *Süptitz et al. (2012)*, a decrease in XCOMap was associated with decreases in COM and COM velocity. Therefore, it was reasonable to assume that providing full visual support led to decreases in the mean value of COM and mean COM velocity even when walking under treadmill perturbations, compared to not providing full visual support from an overall perspective. As expected, an increase in the mean of MOSap was found in the Slip condition compared to the Norm condition. The slowed-down and sped-up treadmill settings could explain this increase in the mean of MOSap, causing the visual system to consistently and rapidly determine self-orientation in relation to the environment step by step. Accordingly, a reduction in the mean value of XCOMap could be considered a reasonable assumption for stabilizing the visual track (*Anson et al., 2014*) and thus increasing the mean value of MOSap.

It may be interesting to ask whether an increase in MOS reflects a ''real'' more stable gait. *Peebles et al. (2017)* found that patients with multiple sclerosis and gait impairments increased MOSap than controls during normal walking. *Stegemöller et al. (2012)* also reported that Parkinson's patients increased MOSap during obstacle negotiation than controls. According to the above-mentioned results, it appeared that the pathological group had a more stable gait (an increase in MOS value) than the control group. As a result, the observations seemed to contradict the general understanding of MOS. Instead, the increase or decrease in MOS values required for each individual task may vary depending on its requirements. On one hand, it may be necessary, for example, to decrease MOSap in order to maintain overall stability while walking on an oscillating platform (*McAndrew Young, Wilken & Dingwell, 2012*). On the other hand, the MOSap could, however, be increased to maintain overall stability when walking on a treadmill, which was perturbed quasi-randomly with full visual support as part of the current study. An ''increase'' or ''decrease'' in MOS value may, in fact, be representative of the strategies necessary to maintain a stable gait at that particular moment (*Curtze, Buurke & McCrum, 2023*) or a specific locomotor task.

## Walking under quasi-random treadmill perturbation with or without full vision support increased MOSml

Was there a reason why MOSml increased in response to a quasi-random treadmill perturbation with or without full vision support? An increase MOSml has been shown in patients with stroke (*Hak et al., 2015*), in patients with spinal cord injury (*Day et al., 2012*), in patients with multiple sclerosis (*Peebles et al., 2017*), in fallers with multiple sclerosis (*Peebles et al., 2017*), in patients with spinal deformity (*Simon et al., 2017*), and in patients with hereditary spastic paraparesis (*van Vugt et al., 2019*) compared to healthy controls during normal walking. Also, for patients with unilateral transtibial amputees, an increase

in MOSml was found in the prosthetic limb compared to the sound limb (*Gates et al., 2013*; *Beltran, Dingwell & Wilken, 2014*; *Brandt et al., 2019*; *Major, Stine & Gard, 2013*). It should be noted that in most of the studies mentioned above, the MOSap either did not differ significantly between patients and controls or was not assessed. Was there a reason why all researchers were interested in the MOSml but not the MOSap? A simple two-legged bipedal toy (*McGeer, 1990*) can answer this question that walking requires less control in the AP direction (passive management) but needed active controls in the ML direction (active control). A similar observation was made that sideway falls were particularly common in older adults; thus, an active adjustment MOSml (*Bruijn & Van Dieën, 2018*) was demanded. It was possible that the increase in MOSml could be attributed to the fact that the overall MOS was affected in both the AP and ML directions as a result of implementing treadmill perturbations in the AP direction. This might also suggest the need for active control to maintain overall stability in the ML direction relative to the Norm direction. Unpredictably, no differences in MOSml between Slip and Slip_VisualBlocked conditions were found, inferring that similar type and intensity of treadmill perturbations in the AP direction might induce similar changes in the mean value of MOSml regardless of the full vision support was provided or not. In contrast, the story was different when investigating the effect of full visual support during walking under quasi-random treadmill perturbations on MOSap and MOSml variabilities step-to-step adjustments in the MOS. The following section provides more details.

## Walking under quasi-random treadmill perturbation with/without full vision support increased MOS variability in the AP and ML directions

Walking under quasi-random perturbations increased MOS variability in both AP and ML compared to regular walking, regardless of whether full vision support was available. Thus, the increase in MOS variability in the AP and ML directions was most likely explained by the experimental design of this study, which included quasi-randomly slowed-down and sped-up treadmill speeds as perturbations in Slip and Slip_VisionBlocked conditions. It is important to note that while the patterns and intensity of treadmill perturbations were similar between Slip and Slip_VisionBlocked conditions (the average treadmill speed in Slip_VisionBlocked condition was 102% of preferred walking speed and 99.8% of preferred walking speed in Slip condition), significant greater mean values of MOSap and MOSml variabilities were still observed in Slip_VisionBlocked than in Slip condition, suggesting a greater degree of adjustments in MOS step by step. There may be a sensory weighting process responsible for this increase in variability (*Winter, 2009*). In light of this sensory weighting process, it has been suggested that gait stability required the combined input of the visual, somatosensory, and vestibular systems (*Herdman, Schubert & Tusa, 2001*; *Peterka, 2002*). Therefore, each sensory channel may be assigned a specific weight. The central nervous system then derived information from different sensory inputs so that it may respond appropriately to different environments. Therefore, if one sensory system was unreliable or malfunctioned (visual system in this study), other reliable sensory systems were heavily weighed to ensure that the central nervous system maintained regular walking patterns. The tradeoff might be the increase in variabilities (*Chien et al., 2014*). Thus, the

reduction in visibility in the Slip_VisionBlocked condition led to the increase in the mean value of MOS variability, which might be attributed to resolving the sensory conflicts from the treadmill perturbations and visual system compared to other conditions.

## Using dominant leg to control the MOS in the ML direction

In healthy young adults, it has been shown that measuring MOSap on one leg or a mean value of MOSap from both legs was sufficient to identify the strategy involving walking at different speeds (*Süptitz et al., 2012*). In spite of this, there was one fundamental limitation to this previous study: MOSml was not discussed. According to *McAndrew Young & Dingwell (2012)*, platform and visual oscillations were associated with asymmetric MOS, suggesting that young adults used the dominant leg for controlling gait stability. The present study found asymmetric MOS while walking on treadmill perturbations without full vision support. Furthermore, the symmetric index was more remarkable in favor of the dominant leg, suggesting that these healthy young adults first attempted to use the dominant leg for challenging locomotor tasks, particularly in the medial-lateral direction. Similarly, *Polk, Stumpf & Rosengren*'s study (*2017*) found that when healthy young adults were instructed to walk with feet laterally rotated compared to normal walking, the asymmetric gait increased by increasing the medial-lateral peak forces and impulses in the dominant leg.

## Limitations

There were a couple of limitations in the current study, as follows:

- The information on each different perturbation may be lost by averaging the outcome parameters over a condition since opposite-direction perturbations may have opposite effects: This study's primary objective was to identify the overall balance control under different sensory-conflicted conditions. Examining how different directional quasi-random perturbations would affect MOS would be interesting. For the current experimental design, however, comparing mean MOS values within a condition among different perturbations may not be feasible. Firstly, there were 17 types of perturbations in one condition, and comparing these 17 types of perturbations even within one condition might increase the statistical errors. If $p < 0.05$ was assumed to be significant, the number of multiple comparisons might reach 136. Thus, the statistical errors might be inevitable. Secondly, these perturbations were not given the same sequence. For instance, an acceleration-type perturbation was not always followed by a deceleration-type perturbation; thus, the perception of participants of each perturbation may be different. Therefore, this study provided the variability measure to overcome this limitation. If the future study aims to investigate the effect of each different type of perturbation on MOS while walking with/without full vision support, the experimental design needed to be re-developed, like Buurke et al.'s study (2019).
- Another limitation was the asymmetry issue about how to identify the number of perturbations on each leg. The research question needed to be clarified in advance to discuss this issue. Based on the previous study about treadmill-induced slip, each perturbation has discretely occurred. In other words, before or after each perturbation, a couple of steps were used to identify the pre- and post-recovery from the slip. Thus,

for data analysis, each MOS per gait cycle during tripping could be clearly picked to answer the research question like "evaluate which types of external perturbations affect the gait pattern the most in terms of ML and AP MoS per each leg" (*Roeles et al., 2018*). However, this study aimed to understand the overall changes in the MOS of each leg under quasi-random treadmill perturbation. Therefore, the experimental paradigm followed a previous study that also investigated the overall MOS of each leg under different types of sensory perturbations (*McAndrew Young & Dingwell, 2012*). Indeed, it was theoretically and technologically possible to identify the MOS of each leg in 17 perturbations. However, it was not feasible to compare the outcomes between two different types of perturbations with/without vision support because the treadmill's speed varied in different time periods.

- The number of steps within two minutes for each participant was different: Although one solution could be made to apply the fixed number of steps for all participants, this solution might raise another problem: the effect of perturbations between two conditions cannot be compared because the total amplitudes and frequencies of perturbations were not equal, particularly in this study.

- This study attempted to understand the instantaneous controls in margins of stability under different sensory-conflicted conditions but not the adapted controls (motor learning) through multiple perturbations: *Cech & Suzanne (2012)* described that "motor control is the physiological process whereby motor development occurs, and motor learning allows motor development to occur systematically, resulting in a permanent change in motor behavior due to experience" in their book. Thus, motor control must be understood in advance to achieve effective motor learning (training). Therefore, this study limited the trial numbers to minimize motor learning. Indeed, while these young adults walked on the treadmill perturbation, the participants' expectations of upcoming perturbations were evitable because they all read the informed consent. Also, step-to-step motor learning may have occurred. This study applied two-minute mandatory rests between trials (mentioned below), randomized the perturbations in amplitude and frequency domains, and limited the trial numbers to minimize this learning between trials.

- Washout period: A two-minute mandatory rest was provided between trials in the present study and was attempted to washout the learning effects between different conditions. This period was based on observations derived from the split-belt treadmill gait adaptation, in which participants walked at different speeds on each leg induced by the split-belt treadmill. The motor leaning was deadapted within approximately 50 strides (around one minute) in washout conditions following a ten-minute awkward gait adaptation (*Reisman, Block & Bastian, 2005*). However, the limitation was whether a two-minute rest was enough to eliminate the learning effect. This present study didn't provide sufficient, direct evidence to support this claim. Future studies should include another overground walking trial between conditions to identify whether the learning effect between conditions still exists. It is important to note that the present study utilized a crossover experimental design to counterbalance this limitation.

- Preference of calculation of MOS: *Curtze, Buurke & McCrum (2023)* suggested that COP should be the preferred and accurate reference point for the calculation of the MOS during walking. Indeed, this study agreed with the point of view of *Curtze, Buurke & McCrum (2023)*. However, the limitation was that calculating the center of pressure during treadmill walking required the expensive instrumental treadmill, and not so many small laboratories could afford this treadmill. As a result, using the heel marker as a reference point may be a practical alternative method for calculating the MOS since this method (heel marker) has been widely used in many studies. (*Hak et al., 2015*; *Fallahtafti et al., 2021*; *Lu, Xie & Chien, 2022*; *Rijken et al., 2015*; *van Vugt et al., 2019*).
- A locomotor sensory organization test #2 condition, which is walking at a fixed preferred walking speed without full vision support, was absent: In such a condition, the function of the somatosensory system could be measured; however, a study already demonstrated that the somatosensory system was barely affected after a long space travel (*Shishkin et al., 2023*). This was the rationale that this study did not consider this condition and may be included in future studies.

## CONCLUSION

The following points summarize the conclusions for the future usability of this paradigm for space missions:

- Walking on quasi-random treadmill perturbations with or without full vision support increased or decreased the mean value of MOSap, respectively.
- Walking on quasi-random treadmill perturbations increased the mean value of MOSml regardless of whether the visual support was provided, inferring that similar treadmill perturbations in the AP direction might have similar effects on the mean value of MOSml.
- Walking on quasi-random treadmill perturbations increased the mean value of MOS variabilities in both AP and ML directions. Moreover, without full visual support (Slip_VisionVlocked), the mean value of MOS variabilities in both AP and ML were greater than other conditions (Norm and Slip), indicating that less sensory system available led to greater variabilities (frequently step-to-step adjustments).
- Providing the quasi-random treadmill perturbations with full vision support increased the MOS in both AP and ML directions to maintain overall stability, suggesting that while walking on a moving surface with vision, the controls in the overall stability were similar in AP and ML direction.
- While walking in a challenging sensory-conflicted condition, healthy young adults tended to use the dominant rather than the non-dominant leg to maintain overall stability.

## ACKNOWLEDGEMENTS

It is with great pleasure that we thank all participants who contributed to this study and thank the editor and reviewers for taking the time to read through our drafts and provide delightful comments to make this manuscript as scientifically sound as possible. This

study was an extended analysis of JC's doctoral dissertation. Data was collected at the University of Nebraska Omaha Department of Biomechanics. The authors sincerely thank the generosity of the Department of Biomechanics for allowing JC to use the equipment to complete his doctoral dissertation. In particular, JC would like to express his gratitude for the years of mentoring provided by his mentors, Dr. Nick Stergiou and Dr. Mukul Mukherjee.

### Funding

The Natural Science Foundation of Sichuan Province No. 2022NSFSC1512 supported Zhuo Wang in completing writing the content of this study. The NASA EPSCoR NNX11AM06A supported this original data collection and analysis for the corresponding author, Jung Hung Chien. The funders had no role in study design, data collection and analysis, decision to publish, or preparation of the manuscript.

### Grant Disclosures

The following grant information was disclosed by the authors:
The Natural Science Foundation of Sichuan Province: 2022NSFSC1512.
The NASA EPSCoR NNX11AM06A.

### Competing Interests

The authors declare there are no competing interests.

### Author Contributions

- Zhuo Wang conceived and designed the experiments, analyzed the data, prepared figures and/or tables, authored or reviewed drafts of the article, and approved the final draft.
- Haoyu Xie conceived and designed the experiments, analyzed the data, prepared figures and/or tables, authored or reviewed drafts of the article, and approved the final draft.
- Jung H Chien conceived and designed the experiments, performed the experiments, analyzed the data, prepared figures and/or tables, authored or reviewed drafts of the article, software programming, and approved the final draft.

### Human Ethics

The following information was supplied relating to ethical approvals (*i.e.,* approving body and any reference numbers):

This study was carried out following the relevant guidelines and regulations and upon approval by the medical ethics committee of the University of Nebraska Medical Center (IRB# 340-10-FB).

### Data Availability

The raw measurements are available in the Supplementary File.

## Supplemental Information

Supplemental information for this article can be found online at http://dx.doi.org/10.7717/peerj.16919#supplemental-information.

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
