# Peer review of "The margin of stability is affected differently when walking under quasi-random treadmill perturbations with or without full visual support"

_PeerJ, doi:10.7717/peerj.16919_

## Round 0.1 · original submission · Major Revisions

Your paper has been reviewed by two expert reviewers. You will find that they have made substantive comments regarding the clarity of presentation as well as the research methodology. Please consider these comments carefully and amend your paper accordingly.

Reviewer 1 ·

Basic reporting

The writing is difficult to follow in places, often due to an unusual choice of words or unclear phrasing, combined with an unclear logical structure and flow (especially in the introduction). A few examples of this from the introduction are listed below. I would recommend having someone with knowledge of the common terms in this field of research proof-read the manuscript and that the authors try to rewrite the introduction with a clearer logical structure. Unfortunately, many of the same issues arise in the discussion section.
Lines 57-58: “have been widely used for one primary purpose to evaluate the patterns of motor control”
Line 60: “these different patterns of response”
In the above two sentences, it is unclear what “patterns” really refers to.

Line 61: “specific diseases can be identified early, such as predicting the likelihood of falls”
Likelihood of falls is not a disease.

Lines 67-70: “These alternations in gait characteristics allow more sensory systems to get involved in exploring environments and further reduce the sensation of uncertainty from environments for balance control.”
The meaning of “get involved”, “exploring environments” “sensation of uncertainty from environments” are all quite unclear here.

Line 77: “Additionally, most treadmill-induced perturbations lead to a single…” I don’t understand the function of the word “Additionally” here since the previous section discussed visual perturbations.

Line 79 “counterbalanced behaviors” is unclear.

Line 81 “Indeed” seems to have been misused here, since the previous sentence doesn’t really have consequence for the next sentence.

Lines 83-87: It is unclear to me why it is a limitation that people can quickly adapt and improve their responses to these perturbations. While the authors are correct that expectation can play a role, there are many studies showing reactive improvements, independent of predictive changes. Many other studies also ensure that the washout phase between perturbations is sufficient to let MoS values return to baseline levels before applying another perturbation, so that overt proactive adaptations in stability are minimised.

Line 85: Should be “healthy” participants.

Line 92: “The results reveal…” It is unclear what this refers to – the previous sentence’s reference to the 2021 study or the current sentence’s reference to the 1997 study?

Line 93: “.., which indicates a relatively unstable gait pattern, mainly for falls” I don’t understand the “mainly” part here.
Lines 94-98: “Can these alternations in gait variabilities represent gait instability? Bruijn et al. (2018) provide an answer that the measure of the gait variability is for balance, which is defined as a sum of instant dynamic equilibrium over time. However, stability is restoring balance without resorting to altering a base of support (BOS) followed by an externally imposed perturbation (Pai & Bhatt, 2007).” My apologies but I can neither follow the writing or the logical organisation of this passage. Reading it word-for-word, I would say that the second sentence is incorrect, since if a person takes a balance recovery step, altering their base of support, to prevent a fall, then this would deem the person to be stable.

Lines 100-102: Change “instant” to “instantaneous”.

Lines 109-110: “Interestingly,..” I don’t follow why this specific result should be interesting to the reader. Please more clearly indicate why this is interesting for readers and why, in the context of the current study, this result is being highlighted in the introduction.

Lines 115-117: “Interestingly,…” Here the topic switches abruptly from visual perturbations to vestibular perturbations with little link or explanation why this should be interesting to the reader or for the context of the study.

Lines 120-122: “the abovementioned control mechanism of stability” I don’t understand what control mechanism specifically has been discussed or referred to here. “..might help physicians/physical therapists to develop a future sensorimotor training protocol.” From the previous text, it is very unclear what this proposal might be based on.

Regarding figure 1, I really struggle to understand what is shown without reference to the text. The arrows on the treadmill surface seem almost identical even though there are differences and the group task boxes are unclear. I actually think I understood the setup better from just reading the text and I am not sure that the image is needed. Videos of the tasks would be very helpful as an appendix.
Regarding figure 2, I would recommend more closely sticking to Hof’s original nomenclature. For example, the extrapolated centre of mass, according to Hof, does not have a height and is a point on the ground. On the right side, the sketch of the leg makes it seem like the leg is swinging, rather than placed on the ground, so I would recommend redrawing this. It is also unclear why the mediolateral BoS velocity is included in the MoS ML calculation.
Regarding raw data, I only find an excel sheet of individual processed data, not true raw motion capture data. I am not sure what level of data should be provided according to PeerJ policies.

Experimental design

As mentioned above, the writing makes it very difficult to understand what the rationale for the current study is. Many different types of perturbations are mentioned with very little logical flow or structure, meaning that I cannot fully understand the research question being addressed. Hopefully with revisions in the writing and structure of the introduction, this will be clearer.

One major issue I have is that it is difficult to determine the authors’ intention with the perturbation protocol, since all sorts of different types of perturbations were discussed in the introduction. The MoS has typically been used to evaluate mechanical stability following acute perturbations or during ongoing perturbations. But here it seems that the protocol is something in between with frequent changes in belt speed at an undisclosed acceleration (note that the changes in absolute belt speed are also much less than those typically used in the simulated trip and slip experiments discusses in the introduction). This also seems to be applied in multiple directions, which will mean that it is likely that balance disturbances in multiple directions will be included in the data. Again, it is not quite clear to me, but it seems like the authors have taken the average MoS values from all steps in the trials. This probably results in some averaging out of distinctive responses, and I am not sure what can really be determined from such averaged values during a multi-perturbation trial. Regarding the asymmetry issue, it is also unclear how the number of perturbations per leg has been controlled. Can the authors revise their text to make their rationale and perturbation protocol clearer and also explain and justify their analysis decisions regarding the MoS?

Lines 148-153: The writing makes this part difficult to follow and it is unclear how two different calculations for statistical power have been combined. Please more clearly report the effect size calculation and all of the parameters used, the type of analysis for which the calculation was performed, the precise source and justification of the expected effect size and the error levels. It would also be helpful to include a screenshot from G*Power in an appendix or in the response to this comment.

Lines 244-246: “However, in the current study, the modified version was used based on Suptitz et al.’s (2012), Fallahtafti et al.’s (2021), and Lu et al.’s (2022) studies, which added the component of the treadmill speed.” I don’t think Fallahtafti and Lu should be cited here or in line 255 – they did not add anything to the methodology here and there are many earlier studies that have used this method. Süptitz (2012) was indeed the first study to apply this correct, but it is also not really true to say that this is a modified method, since the calculation is the same and it simply accounts for the relative velocity.

Lines 258-259: “Therefore, if the dynamic stability was considered relatively stable in the anterior-posterior direction, the values of MOSap should be negative.” This seems to be incorrect. If you use the formula according to Hof and subtract the XCoM from the BoS (BoS-XCoM) then negative values indicate instability (the XCoM is outside the BoS). In general, the following sentences (Lines 261-270) give a very confusing explanation of the calculation and the meaning of the negative values. For example, what the authors write for AP is not consistent with what they write for ML, in which they (correctly, according to Hof) state that positive values indicate stability. In the final sentence of these lines, they suggest that instability is not meant as losing balance, but the reasoning is not clear why. Here it would be important to note that a negative MoS at a specific instance indicates mechanical instability at that instant, but does not necessarily say anything about the general balance capacity of the person during the entire duration of a specific task. This section should be carefully and precisely rewritten to make both the calculation and explanations clear, consistent and mechanically sound. Similar issues are found in the discussion (Lines 362-363, 378-382, 390-394).

Lines 275-276: Using the standard deviation is not valid if the same number of steps per trial/participant are not used, since the SD is affected by the number of data points. Rather than calculating this for all steps during a two-minute period, consider calculating SD for the final 100 steps of each trial, or something similar.

Lines 286-298: Given the lack of clarity on the specific issues being addressed and the relatively high number of variables being tested, there needs to be more clarification of the specific hypotheses being tested, their rationale and how these relate to primary and secondary variables.

Lines 471-472: Many studies in the literature use walking speed as a covariate in statistical analyses. Please remove this statement.

Lines 195-196: “The trials were limited because this study focused on the motor responses to unexpected perturbations”. In this type of protocol with frequent perturbations, how can the authors exclude any expectation?

Lines 227-229: “Each condition lasted for two minutes, and a two-minute mandatory rest was provided between trials for participants to eliminate any learning effects from the prior trial.” The authors state that this was based on a split-belt study, but there are many studies with trip and slip perturbations that consistently show very long-lasting retention of adaptations to such perturbations (months to years) so I find it difficult to see how only two minutes would be enough to eliminate learning effects. Do the authors have data that can support this?

Validity of the findings

It is difficult to evaluate the results and conclusions with the lack of clarity in the introduction and methods. With revisions to these sections, it hopefully becomes clearer what the results really represent and imply.

Lines 339-340: It is very surprising that no significant effects of the perturbations were found for the gait parameters when significant effects were found for the MoS parameters. Since step length and width should closely correspond to the BoS component of the MoS calculation, I would expect some differences to be found. For this reason, I recommend reporting also the BoS and XCoM values separately to help determine how these change and contribute to the MoS changes observed.

Additional comments

Lines 71-72: Harness systems and other body weight support systems are common features in many clinical and rehabilitation settings. I do not believe that this is a major barrier and is at most only a perceived barrier.

Line 74: Please explain the difference between the blindfolded and blackout glasses methods.

Line 102: The velocity of the treadmill belt is not specifically a critical, distinct component of Hof’s margin of stability. It is simply a correction for relative velocity necessary when assessing the MoS while walking on a treadmill. It is also not needed in the ML direction for instance. It is also not strictly the location of the BoS that is necessary, but the location of the boundary of the BoS.

Lines 103-108: It is unclear why different MoS results in very different tasks is controversial. Different tasks will have different effects on participants. Why is this controversial? Calculation methods may also play a role in these differences. Can the authors clarify the purpose/meaning of this passage?

Lines 127-129: The hypothesis of the perturbations inducing an asymmetric gait pattern is not explained or justified in the introduction.

In general, it is unclear to me why the perturbation condition has been given the abbreviation “SLIP” since it appears that both increases and decreases in belt speed are applied during the same trial and the changes in belt speed are typically much lower than previous studies simulating slips with treadmill belt perturbations.

Reviewer 2 ·

Basic reporting

Overall, the English language is clear and well written, although in numerous sentences it appears a word is missing. E.g. check sentences on r.43, r.48, r.90. Also the choice of terminology shoud be done more careful as sometimes language is to informal e.g. "unexpectedly enough induced treadmill perturbation" r.373 (see also next comment)

Your introduction needs clarification of specific concepts and terminology you are using (see below, also annotated comments in the PDF)
In the introduction many complex concepts are introduced, such as "relative- unstable gait" (r.81), "balance" (r.95), "stability" (r.96) but they are not well defined or, when a definition is provided, it does not always match with how the concept is described in the papers that are referred to. I would like to suggest to thoroughly reconsider what concepts are essential and provide definitions that are in line with literature. There is a lot of confusion around the terminology of balance control and stability, and therefore it is essential that the reader grasps your interpretation of these concepts from the beginning. Also the concept of MOS (r.100-103) can be described in more detail, since you only refer to how it is calculated.
Also treadmill perturbations are discussed, but it is not clear to the reader how "treadmill-induced perturbations" are different from "continuous perturbations"

Literature well referenced & relevant.

Structure conforms to PeerJ standards: the research has been approved by the IRB from 04-18-2013 till 04-18-2014 but the period of data collection is not specified in the manuscript. Written informed consent was obtained from all participants prior to data collection (as specified on p.9 r.143)
Requirements for human subject research were met: (1) approval from IRB was obtained, which is explicitated in the manuscript on p.9 r.140 (including approval reference number); (2) written informed consent was obtained from all participants and an empty consent form is available in the supplementary material; (3) identifiable information is not available in the manuscript or supplementary data

Figures are relevant, high quality, well labelled & described.
Raw data are provided with sufficient detail.

Experimental design

The introduction lacks background information and does not clearly identify a knowledge gap. Specifically, this information should be elaborated on lines 117 - 122. It is not sufficiently clear here, what specific perturbations need to be investigated, what control mechanisms are referred to and why suddenly "symmetry" is relevant (see also my comments in the annotated PDF).
It is not clear to me what is the research question. Only in the discussion, some hypotheses are put forward (r.366) that might be better introduced already beforehand.

A strong point is that you provide a sample size justification. Marker placement and details of measurement are well described. Gold standard equipment for movement analysis is used.
You should provide some more information on what the treadmill perturbation actually is on r. 185-188 (see also my comment in the PDF)
Regarding the statistical analysis, I am not clear why for part of the outcome variables (MOS related measures), preferred walking speed is not added as a co-variate while for other outcomes it is. If I understand correctly, it is not the actual walking speed but the preferred speed (which is determined beforehand and is the speed of the norm condition), that is used. Gait parameters indeed depend on walking speed, it is a complex relation and adding walking speed as a co-variate makes sense when you are comparing groups or conditions where speed might differ. But in this case, you have repeated measures and, while speed might differ between participants, it does not between your conditions. So why would it be a covariate? Is the lack of significant results in the ANCOVA not just a reflection of lack of statistical power when adding a covariate?

Methods not always described with sufficient detail & information to replicate:
The way the MOSap is interpreted is confusing. Consider clarifying r.260-263.
Since in the introduction, "a relatively-unstable gait" is not clearly defined, it is not clear to me why you say that a larger MOSap (larger negative values) and a larger MOSml (larger positive values) reflect an unstable gait.
You should describe how you treat your main outcome parameters over the entire trial, where they averaged over the trial? But then how is this related to the induced perturbation that lasts 5 - 10s? From figure 1 I understand that the belt is constantly accelerating and decelerating, but I would expect a different response in step length and MOSap in case of an acceleration than in case of a deceleration. So how do you take this into account in your data analysis? This information should be added before the statistical analysis is described (r.285)

Validity of the findings

Impact and novelty should be better assessed:
You are comparing three condition: walking under normal light conditions and at constant speed, walking under normal light conditions but with quasi-random perturbations of the treadmill and walking under reduced light conditions with quasi-random perturbations of the treadmill. If you want to completely understand the sensorimotor control of gait stability, and unravel the importance of vision in relation to other perturbations, a condition is missing i.e. walking under reduced light at a constant speed. You should consider this in the discussion as a limitation. Also, you cannot entirely compare your results (with 2 perturbations) to the studies that only use a visual perturbation, as you do on r.357-366, without acknowledging this.
You are generating confusion with the summary of findings, check r.348-353
In the discussion of your results on changes in MOS under reduced light and treadmill perturbations, you put forward 4 different explanations but you should be more specific on how exactly they can explain your findings. (r.357 and further)

Conclusions are not well stated and not limited to supporting results.
In general, the discussion remains superficial. Several theories on control of gait stability and balance control in general are briefly described but the link with the findings of the current study remains vague. I would ask the authors to go more in depth in the discussion and provide clear arguments why a certain hypotheses is confirmed or rejected. Also, what do we learn from that research that we do not already know? The novelty could be stressed, or it the aim is to confirm previous hypotheses, this should also be made more clear in the text. Sometimes, a clear hypotheses is provided, but only in the discussion (e.g. r.445), this should be in the introduction.
The implications and conclusions that are provided are not supported by the research results and should be carefully reconsidered (see also my comments in the PDF).

Additional comments

You might reconsider the way you refer to the different conditions, to avoid confusion. Norm and Slip are clear, but SlipVision is sometimes confusing. It might be better to refer to SlipDark (or something similar), so the reader immediately understands this is the reduced light condition. Also "without full vision" seems a bit elaborate to refer to a reduced light condition. It is always more clear to state what something is, instead of what it is not.

Annotated reviews are not available for download in order to protect the identity of reviewers who chose to remain anonymous.

---

## Round 0.2 · Minor Revisions

The reviewers are happy with the revisions made but have identified some remaining concerns. Specifically it is important address the issues on the interpretation of the MoS as mentioned by reviewer 1 as well as to specify in the text that and how treadmill speed was taken into account. Reviewer 2 comments on averaging over perturbations in different directions and I would recommend to account for perturbation direction in your analysis.

Reviewer 1 ·

Basic reporting

I appreciate the authors reworking their introduction at great length. I think the personal rationale of the authors, in their support from NASA, is now clearer. Though I do think the first sections of the introduction are now quite lengthy and could potentially be condensed.
My previous comments on the figures have been addressed.
The aims and hypotheses are also now much clearer.

Experimental design

Regarding the wash out time between trials, the authors now indicate that the 2 minutes was expected to prevent learning effects influencing the following block. They support this by referring to the 30-90 seconds washout used by McCrum et al (2018). However, in McCrum’s study, this was the washout time between perturbations, not trials. The 30-90 seconds is sufficient time for the acute effects of one perturbation to dissipate and for gait to return to baseline MoS values during unperturbed gait. It is not sufficient time between trials to reduce adaptations to repeated perturbations. That being said, the authors have counterbalanced the order of the perturbations trials, which may address this issue to some extent.
Regarding the MoS, the original publication introducing the concept is Hof et al (2005) Journal of Biomechanics. Not the 2008 paper as written in the methods.
In addition, it is worth highlighting that for consistent forward walking, the MoS AP should be negative. Otherwise, efficient forward progression is less achievable. This is part of the issue with using the toe marker as the base of support boundary – the COP at heelstrike is not at the toe, it is at the heel. I recommend the authors have a look at this recent paper addressing some of these common issues with applying the MoS: doi: 10.31219/osf.io/nym5w
While it was mentioned in the previous manuscript and in the comments and responses, it is now unclear in the manuscript if a correction for treadmill velocity was conducted for the XcoM position.
Regarding the discussion section on how to interpret increased MoS, particularly in patient groups in which we expect instability, I again encourage the authors to have a look at the recent paper mentioned above where this is discussed based on previous suggestions, also from Hof. One related issue is that the MoS does not measure gait stability in a global sense, only the mechanical stability at a specific instance. Patients or people with deficits can passively take advantage of this by increasing their instantaneous mechanical stability to compensate for their less effective control.

Validity of the findings

In general, I don’t think the conclusions about greater reliance on vestibular function are supported by the data. Certainly, the conclusions that this sort of training can enhance vestibular function is not supported by the data in the current study.
Regarding the McAndrew and McAndrew-Young studies, it is worth highlighting that the MoS AP values in that study are not reliable, since the authors of those studies erroneously did not correct their XcoM values for the treadmill speed. As a result, any discussion or comparison with their MoS AP values can lead to incorrect conclusions. I would therefore suggest that the authors reconsider their discussion of these findings in both the introduction and discussion, particularly in the discussion where many comparisons of the values are made.

Additional comments

I thank the authors for their thorough revision of the article. Many of my previous comments have been addressed. I hope my comments on the remaining issues and clear and useful for further revisions the manuscript.

Reviewer 2 ·

Basic reporting

The language has improved and the text is now much easier to read. Also the background is more relevant and there is a clear rationale for the study. You provide a definition of stability (r.169) which is a core concept of your study. The research goal is clearly stated and there is a hypothesis formulated.

Some minor language suggestions (rule numbers are taken from the document with track changes):
r.181: … to prevent the human walking from falling away  Do you mean “falling to the side”?
r. 243 – 245: Here you present your hypothesis, but it is not entirely clear to me what is the difference between hypothesis 1) quasi random perturbations without full vision and hypothesis 2) without full vision support
r.625 – 700: This is a large block of text. To improve readability, I would suggest to add paragraphs when you introduce a new concept or idea. E.g. r.670 introduces a new idea to explain the decrease in MOS ap. Here you could start a new paragraph.
r.676 – 677: Conversely, a smaller MOS can be a shorter boundary of BOS and farther XCOM regard to COM. Thus, a smaller MOS indicated a closer distance between the boundary of the BOS and the XCOM – Here you are giving the same message twice.

Experimental design

Regarding experimental design, there are clear in- and exclusion criteria for the participants. However, can you clarify whether participants were recruited from the general population or from a specific group? This information is relevant to draw conclusions on generalizability of the results.
The study design appears adequate and is clarified in figure 1B.

Some suggestions:
r.503: Please add whether you refer to the toe marker of the leading or the trailing leg. Also add this information on r. 528, M5 of leading or trailing leg?
r. 563: Can you provide some indications on how to interpret eta squared (small, medium, large effect)?

Validity of the findings

Regarding the validity of the findings, I still have some concern about the way the data is processed. By averaging your outcome parameters over the entire trial, you might loose a lot of information as opposite perturbations might have opposite effects. I do agree that this should be reflected in the increased variability (which you do observe) but interpreting the mechanics or the strategies used in response to the perturbations becomes less straightforward. Therefore, you should be more careful not to overinterpret the significance you find.

Some comments on the interpretation of the findings in the discussion:
r. 665: “… an enlargement in the amplitude of perturbations or manipulations might be necessary” – What do you mean by enlargement? Larger compared to what? And are you referring to visual perturbations or platform perturbations?
r.669: What do you mean by unexpected enough?
r.691 – 693: “The vestibular system needed to provide a sense of stability and information about body position; therefore, a quick trunk movement was required to allow rapid compensatory movements in response to externally generated perturbations (Cullen, 2012)”  What do you try to explain here? The vestibular system does not give information on trunk movement but on head position and acceleration. Would you assume that the head is fixed on the trunk? This is not what the review of Cullen is about.
R.824 – 827: “This resulted in the MOSVml and MOSVap increasing due to the extra time required to integrate the environmental information into the reweighting process.”  Another explanation can be that increased variability is the result of the required step to step adaptations.
r.849: How does an asymmetric MOS tell you that someone is heavily relying on vestibular information? The reason you only find an asymmetric MOS while walking on the treadmill with perturbations and without full visual support, might just be because it is the most challenging condition.
r.982: “It appeared that walking in such a sensory-conflicted condition (Slip_VisionBlocked, where both the visual and somatosensory/proprioceptive systems were perturbed simultaneously, Chien et al., 2014) heavily relied on vestibular information . Thus, training under such circumstances may enhance vestibular function, which has been altered primarily in different gravities to resolve gait instability.”  I do not agree that this is a conclusion of your study, because you cannot directly measure the use of sensory information. This is an assumption.
R. 989: “The asymmetric gait pattern needed to be paid extra concerns …”  What do you mean by this?
r. 993 – 995: “Also, this synthetic paradigm can serve as a pre-and on-board training tool that provides astronauts with sufficient sensorimotor challenges and helps them improve their motor response adaptability as they prepare for various gravitational transitions .”  You cannot conclude this will improve gait stability before you indeed do the training in a (simulated) reduced gravity environment.

Additional comments

No additional comments

---

## Round 0.3 · Major Revisions

Reviewer 1 has asked to derive corrections made in the XcoM calculation for treadmill speed for the studies cited rather than just conclude that this was not reported In addition some references appeared incorrect. It is important to carefully consider the issues regarding the validity of the interpretation of the results identified by reviewer 2. These issues were introduced in this revised version. They can be overcome by a revision without additional analyses of the data collected or new analyses, but will require a major revision of in particular the discussion section.

Reviewer 1 ·

Basic reporting

The authors have addressed my previous comments. Here are some additional minor comments:

Line 126 in the track change document: Hof (2005) should be corrected to Hof et al. (2005).

Lines 140-142 in the track change document: "Also, a negative MOSap would represent an interruption of forward progress and is related to a minimal impulse for returning the body to balance (Curtze et al., 2023; Lu et al., 2022; Fallahtafti et al., 2021). " - This is incorrect - a negative value does not represent an interruption to forward progress. This sentence in general though is not necessary for the section it is placed within, so I would suggest deleting it.

Lines 176-183: While the methods sections do not reporting making a correction in the XcoM calculation for treadmill speed, it is not possible to conclude from the figures easily if this was done or not for Madehkhaksar et al., 2018 and Roeles et al., 2017, whereas it is visible in the figures for McAndrew-Young et al. 2012. I would suggest the authors attempt to clarify if those other two studies made the correction or not before stating this so categorically.

Experimental design

The authors have addressed my previous comments. Two additional comments:

Lines 366 - 383: While this text provides a lot of information, it is not relevant to the reader for understanding your study and the presentation of the two methods may be confusing. I would recommend removing this.

Lines 389-390 are confusing. Mentioning greater and smaller in the context of a negative value is problematic. I suggest refining this part (also for the ML) to only indicate the direction of calculation and the chosen BOS boundaries. Please also remove the references to Fallahtafti and van Meulen in this section - they are not suitable.

Validity of the findings

The authors have addressed my previous comments.

Additional comments

Line 880: Just because a method has been widely used, does not make it a good alternative. I suggest altering the wording here to "a practical alternative".

Reviewer 2 ·

Basic reporting

Figure 4 and 5
Legend and figure titles do not match. The legend refers to MOSap and MOSml but figure 4B shows the MOSVap (so the coefficient of variation).
Similarly, there seem to be some errors with figure 5 as well.

Experimental design

no comment

Validity of the findings

The discussion is completely different from the previous version, despite that both reviewer #1 and myself gave very specific comments on parts of the discussion and conclusion, warranting some caution regarding overinterpretation of the results.
By completely reworking the discussion, I get more concerned on the validity of your findings than in the previous version (e.g. your interpretations regarding the lack of step to step adjustments – see further)

r. 553-555: “The present study, however, found no significant difference between walking normally and walking under treadmill perturbations without full vision support.” – no significant difference in what?
r. 556: Why is it reasonable to assume that changes in MOS are primarily related to changes in the XCOM? You do not yet convince me of this. I do not see how an average treadmill speed motivates the lack of step to step adjustments after perturbation. Furthermore, treadmill speed was something imposed by the experimenter, not the participant, so how can it explain postural control strategies? Since your second hypothesis (that a smaller COM velocity is to stabilize the trunk), builds further on this assumption, this might also be flawed.
You apply the same reasoning in the paragraph r.633 – 639. So, since the average walking speed is not very different from self-selected speed, you expect that on average there is no effect of step length. But the average speed is largely dependent on the imposed protocol, and that includes both speeding up and slowing down the treadmill, so on average, you do not expect any differences. I feel that the major problem in your reasoning is that you falsely interpret the lack of average effects as a lack of instantaneous postural responses. But your perturbations are direction dependent and so are your outcome parameters, so you cannot say that average responses and instantaneous responses will be the same.

r. 712 – 738: Here you discuss the increased variability in MOS. Be careful not to overinterpret your results. I think a large part of the variability you observe should be attributed to the imposed protocol (randomly speeding up and slowing down the belt) that requires step to step adjustments. You acknowledge this, but only as a secondary explanation, while I think it is probably the main reason for variation in your results. If sensory information would really be important, why is there no difference in variability between the visual conditions?

Additional comments

Introduction
I understand reviewer #1 suggested to make the introduction shorter, but you deleted context (e.g. main aim of the present study)

Discussion:
R.501-504: Your conclusions are confusing, here. Point 1) states that “with full vision support, quasi-random treadmill perturbations reduced the MOSap” and point 2) states that “with full vision support , treadmill perturbations increased both MOSap and MOSml”. Both cannot be true. I suppose you mean “without full vision support” in point 1?
r. 577 – 581: “Although head movements were not recorded in the present study, participants were asked to describe their feelings following the completion of each condition. Participants have described wearing goggles while the treadmill speed is constantly changing, causing them to lose their orientation while walking. It was therefore necessary for participants to maintain a steady upper body position in order to determine where they were.” – How is describing a feeling of unsteadiness a substitute for not having recorded head movements? Furthermore, maintaining a steady body position is motor output, not sensory input. It seems you are mixing up these concepts.
r.704 “ ; the increases in step length caused the greater MOSml in this current study” – Why step length? The paragraph talks about step width.

Limitations
r.790 Thank you for addressing averaging as a limitation. I understand that your study would be underpowered to address all perturbations separately. Nevertheless, I advice you to be more cautious in your interpretations.

---

## Round 0.4 · Minor Revisions

Please address the minor comments that the reviewer has made and also consider language editing of the paper to improve readability.

**Language Note:** The Academic Editor has identified that the English language must be improved. PeerJ can provide language editing services - please contact us at [email protected] for pricing (be sure to provide your manuscript number and title). Alternatively, you should make your own arrangements to improve the language quality and provide details in your response letter. – PeerJ Staff

Reviewer 2 ·

Basic reporting

I suggest that a fluent English-speaking person does a grammar check of the entire manuscript.

Experimental design

no comment

Validity of the findings

no comment

Additional comments

R. 499 (no track changes): "Thus, no significant differences in step length were found between the Norm and Slip_VisionBlocked conditions in the present study." - Replace "Thus" with "Also"

R. 505 - 507 (no track changes): "Accordingly, the decrease in MOSap's mean value was therefore attributed to the increase in XCOMap's mean value after step-to-step adjustments when comparing this Slip_VisionBlocked condition with the Norm condition (combination #5 above). " - Formulate this more as a hypothesis e.g. replace "was therefore attributed to" by "might therefore be attributed to"

R. 528 - 530: "Thus, the increase in the mean value of MOSap in this Slip condition may be attributed to the increase in the mean of value of BOSap but the decrease in the mean of XCOMap compared to Slip_Visionblocked condition." - Please check the grammar of this sentence.

---

## Round 0.5 · Minor Revisions

I have personally assessed the minor revisions made. Overall, the paper is ready for publication.

However, I noticed one important omission. In the sentence: "increaseThe increases in the mean value of MOSap while walking in thisthe Slip condition maycompared to walking in the Slip_VisionBlocked condition in the present study might, therefore, be attributed to the increase in the mean of value of BOSap but the decrease in the mean of XCOMap compared to Slip_Visionblocked condition.(combination #2). " The word not is missing. It should, I think, read: " increaseThe increases in the mean value of MOSap while walking in thisthe Slip condition maycompared to walking in the Slip_VisionBlocked condition in the present study might, therefore, not be attributed to the increase in the mean of value of BOSap but to the decrease in the mean of XCOMap compared to Slip_Visionblocked condition.(combination #2). "

Please make this final remaining correction, after which I will be happy to accept your paper.

---

## Round 0.6 · accepted · Accept

Thanks for making this final correction and for this nice contribution to the literature.